# Ray-Based Analysis of Subcritical Scattering from Buried Target

Yeon-Seong Choo [1,2] , Giyung Choi [3], Keunhwa Lee [3], Sung-Hoon Byun [1,2] and Youngmin Choo [3,*]

1 Department of Ship and Ocean Engineering, Korea University of Science and Technology (UST), Daejeon 34113, Republic of Korea
2 Ocean and Maritime Digital Technology Research Division, Korea Research Institute of Ships and Ocean Engineering (KRISO), Daejeon 34103, Republic of Korea
3 Department of Ocean Systems Engineering, Sejong University, Seoul 05006, Republic of Korea
* Correspondence: ychoo@sejong.ac.kr; Tel.: +81-2-6935-2532

**Abstract:** A ray approach is used to simulate subcritical scattering from a buried target at low-to-high frequencies (100 Hz–15 kHz). A penetrating wave at a subcritical angle decays along the depth at the bottom (i.e., evanescent wave) and propagates horizontally at a subcritical angle-dependent speed lower than the sound speed of the bottom. The corresponding target strength (TS) is distinguished from that of a standard plane wave. Its pattern is asymmetric by the evanescent wave including for symmetric targets and is more complicated owing to the higher wavenumber induced by the lower speed of the evanescent wave. A scattered signal is simulated by considering the features of the penetrating wave with the TS and then verified using the finite element method. In the ray approach, once the TS is computed, a scattered field is efficiently derived with low computational complexity. Strong peaks are observed in the scattered signal via mid-frequency enhancement; however, their amplitudes are less than those yielded by the free-field target owing to the more diminished penetrating waves at higher frequencies. The peaks indicate the possibility of detecting the buried target using a receiver near the target (bistatic sonar) with a broadband source signal that includes low-to-mid frequencies.

**Keywords:** buried target detection; subcritical penetration; ray approach; mid-frequency enhancement; bistatic sonar

## 1. Introduction

It is important to detect objects near the seabed for mine countermeasures. Mines, which are on the seabed when installed, are buried gradually under the sea bottom over time, rendering mine detection more difficult. In particular, it is desirable to detect the buried targets at a long range, while a sound wave from the sonar penetrates the sea bottom surrounding the target at a low grazing angle. In a typical ocean environment with a hard bottom (whose sound speed is higher than that of the water column), the sound wave becomes evanescent at the bottom owing to the low incident grazing angle below the critical angle (subcritical angle). This induces complicated scattered signals that must be analyzed to determine the detection performance of the active sonar systems from the mine.

Numerous studies on target scattering have been performed, where scattered signals are modeled to inspect the physics of target scattering. Additionally, numerous experiments have been conducted to verify the modeling using physics. Man-made targets, including mines, are typically regarded as spherical or cylindrical shells supporting surface elastic waves, which cause the target signal patterns to vary rapidly with frequency. In particular, intensive scattered signals from a spherical shell in the mid-frequency range (mid-frequency enhancement (MFE)) were analyzed via modeling, where elastic waves with specular echo were considered to use the features as indicators for mine detection [1,2]. Cylindrical shell scattering was investigated while prioritizing exceptional elastic waves, such as helical waves propagating on the surface in an oblique direction [3–5].

Scattered signals from targets near the sea bottom exhibit patterns that differ from those for the target in the free field owing to multiple scattering paths (target above sea bottom) or wave penetration into the bottom (target below sea bottom). The effects of the sea bottom on target scattering were examined using simulated scattered signals based on the target burial depth and sea bottom type, and the T-matrix method was adopted for the simulations [6,7]. Exploiting advances in computational power, Li et al. analyzed the main elastic scattering components from a buried spherical shell based on the finite element method (FEM) [8]. However, only the supercritical angle was considered as the incident angle on the sediment. Zampolli et al. efficiently modeled target scattering near the sea bottom using the FEM [9]. For the computational efficiency, an approximate Green's function was incorporated with near-field scattered pressure and normal velocity of axially symmetric target obtained from the FEM in Helmholtz integral. Subsequently, the efficient FEM was adopted to investigate acoustic scattering from cylindrical targets [10,11].

Although the FEM yields an accurate numerical solution, intensive computational power and memory are required for modeling target scattering. Hence, a ray approach was used to model the target scattering above the sea bottom, where the target was treated as a directional point source by a target scattering cross-section and the interference effect from the seabed with multiple paths was interpreted; the results indicated good agreement with the measurement data [12,13].

The physical background of a penetrating wave at the subcritical angle and the corresponding target scattering were investigated [14–16] because an incident wave below the critical angle becomes different in terms of propagation properties after penetrating the sea bottom, and scattered signals from a buried target by the penetrating wave exhibit unique scattering characteristics. Maguer et al. inspected the subcritical penetration into the sea bottom by measuring pressures at a fixed buried depth along various incident angles and then analyzed its physical characteristics via wavenumber integration-based modeling [14]. The penetrating waves at high frequencies were explained based on the bottom roughness, whereas those at low frequencies (5–7 kHz) were described as evanescent waves decaying along the depth. Lucifredi et al. analyzed the corresponding scattered signals [15]. The dominant components for scattering differed based on the frequency, i.e., specular echo and elastic waves dominated the low and high frequencies, respectively. This restricts the ray approach to low-frequency scattering. Simpson et al. measured backscattered signals from a buried target based on incident waves at subcritical and supercritical angles [16]. As expected, the signals scattered by the penetrating waves below the critical angle were weak and decreased rapidly as the frequency increased.

The aim of the current study is to elucidate the mechanisms of buried target scattering at low-to-high frequencies (100 Hz–15 kHz herein) by incident waves at subcritical angles and then analyze the scattered signals in terms of long-range target detection. In Section 2, an analysis of the propagation properties (direction, speed, and decay rate along depth) of penetrating waves at subcritical angles via the FEM is presented, as well as the modeling of the corresponding target strengths (TSs), which are distinguished from those of standard plane waves. Scattered signals were simulated along the frequencies using the ray approach by considering the penetrating waves and TSs, which provides physical insights into the buried target scattering. Based on the simulated scattered signals, the possibility of detecting buried targets was investigated using a bistatic sonar system with a broadband source signal inducing the MFE, as described in Section 3. Finally, Section 4 provides the conclusions of the study.

## 2. Modeling Scattered Signals from Buried Target by Subcritical Incident Wave

A scattered signal from a buried target by a subcritical incident wave was simulated via the FEM using COMSOL Multiphysics® [17] to investigate the active sonar performance for detecting the buried target over a long range. To reduce the computational burden of the simulation, a two-dimensional (2-D) cylindrical shell was used as the target. The numerical environment for the scattering modeling is illustrated in Figure 1.

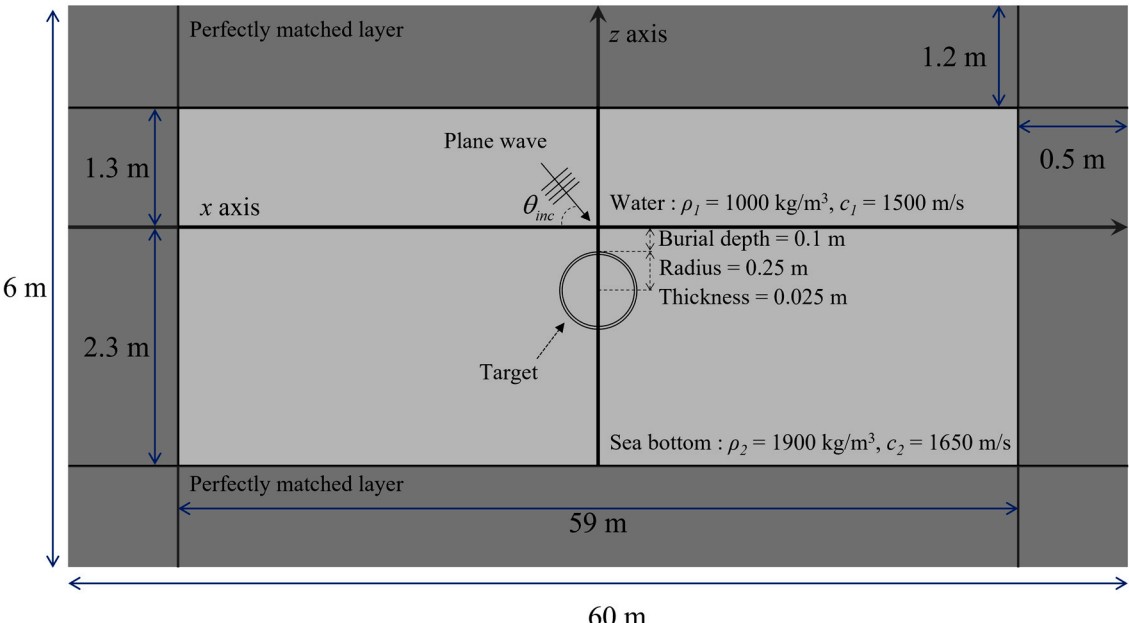

**Figure 1.** Numerical environment for subcritical scattering from buried target using FEM, where incident angle of $\theta_{inc}$ is less than critical angle of $\theta_c$. The width and height of FEM are 60 m and 6 m, respectively, and two fluid mediums with the size of 59 m × 3.6 m are surrounded by a perfectly matched layer (PML). The aluminum shell target, whose density and longitudinal and transverse speeds are 2700 kg/m³, 6148.9 m/s, and 3097.3 m/s, respectively, is fully buried in the sediment and its inside is filled with air. Triangular and quadrilateral meshes are used for two fluid mediums and PML, respectively, and the mesh size is the one-twelfth acoustic wavelength.

The width and height of FEM were 60 m and 6 m, respectively, and two fluid mediums with the size of 59 m × 3.6 m were surrounded by a perfectly matched layer (PML); the two fluid mediums consisted of water and sandy sediment with the heights of 1.3 and 2.3 m and had open boundaries. The sediment was assumed to be sand, but shear properties were not considered. Densities ($\rho_{1,2}$) and sound speeds ($c_{1,2}$) of water (upper) and sandy sediment (upper and lower fluid mediums, respectively) are 1000 kg/m³, 1900 kg/m³, 1500 m/s, and 1650 m/s, respectively. The aluminum shell target, whose density and longitudinal and transverse speeds are 2700 kg/m³, 6148.9 m/s, and 3097.3 m/s, respectively, was fully buried in the sediment and its inside was filled with air. The target center was at the depth of 0.35 m from the water–sediment interface and the target radius and thickness were 0.25 m and 0.025 m, respectively. The incident plane wave from the water penetrated the sediment with a subcritical angle (20° herein), and scattering occurred owing to the buried target illuminated by the penetrating wave. The scattered signals corresponding to the frequency range of 100 Hz–15 kHz was simulated using the FEM with a mesh size of the one-twelfth corresponding acoustic wavelengths. Triangular and quadrilateral meshes were used for the two fluid mediums and PML, respectively. All parameters including the model and mesh sizes were determined through a convergence test; the FEM results were observed according to various model and mesh sizes.

Although the FEM provides an accurate numerical solution with an appropriate mesh size and boundary condition, it is impractical for the modeling of a realistic three-dimensional (3-D) target (particularly at high frequencies used for synthetic aperture sonar and side-scan sonar [18,19]), owing to the high computational time and memory required for computation. Furthermore, the numerical approach does not allow a physical interpretation of the scattered signal, which is crucial for analyzing the active target detection performance.

To simulate the scattered signal from the realistic target, an efficient model for the buried target scattering was developed using the ray approach. It is noteworthy that the

ray approach has been used to model scattered signals from a target in a free field or on the sea bottom by combining wave propagation from the source to the target and from the target to the receiver with a target scattering pattern along scattering angles (i.e., the target scattering amplitude or cross-section) [12,13]. However, to model long-range buried target scattering, the ray approach is restricted to low-frequency regions, and the target scattering at high frequencies is simulated by considering the bottom roughness [18–20]. In this study, the ray approach was extended to model the target scattering at low- and high-frequency regions (100 Hz to 15 kHz) under a perfectly flat bottom by inspecting the wave propagation of incident waves below the critical angle in the sediment and the corresponding target scattering cross-section, thereby allowing the ray-based unified scheme to model scattered signals from buried targets over a long range.

The wave propagation of subcritical incident waves exhibits the following features: (1) The direction of sound is parallel to the boundary between water and the sediment; (2) the wave propagation speeds in the sediment depend on incident angles below the critical angle; (3) the sound becomes evanescent along the depth, and the decay rate depends on the frequency and incident angle. Section 2.1 provides an explanation of the features in detail. Section 2.2 presents an analysis of the target scattering cross-section of the target illuminated by the evanescent wave having the same decaying rate and sound speed as the subcritical penetrating wave (i.e., subcritical scattering from a buried target), based on a comparison with that of a standard plane wave. Section 2.3 presents a unified ray approach for modeling the scattered signals of the long-range buried target based on conclusions from previous subsections.

## 2.1. Incident Waves below Critical Angles

A long-range incident wave from the source can be approximated using a plane wave with a low grazing angle that penetrates the sea bottom at a subcritical angle in an ocean environment involving a hard bottom. Because the wave traveling along the boundary must coincide with the horizontal components of waves in water and the sediment (i.e., $(k_1)_x = (k_2)_x$, where $k_1$ and $k_2$ are the wave vectors of the first (water) and second (sediment) mediums, respectively, and the subscript $x$ indicates the horizontal components), the penetrating wave in the sediment propagates in the direction parallel to the boundary regardless of the incident angles below the critical angle $\theta_c = \cos^{-1}(c_1/c_2)$ ($c_1$ and $c_2$ are sound speeds in water and the sediment, respectively) and becomes evanescent along the depth. The decay rate $\alpha(f)$ is defined as follows:

$$\alpha(f) = \ln\left|\frac{p(z)}{p(z + \Delta z)}\right| / \Delta z, \tag{1}$$

where $p(z)$ is the pressure at depth $z$. Subsequently, the decay rate is expressed analytically as $\alpha(f, \theta_{inc}; c_1, c_2) = \sqrt{(k_1)_x^2 - k_2^2}$ for the penetrating wave below the critical angle, where $k_2$ is the wavenumber of the sediment (i.e., $k_2 = |k_2|$), and $\theta_{inc}$ ($< \theta_c$) is the incident angle of the wave into the sediment [21].

The penetrating wave, which propagates horizontally in the sediment, has a speed of $\frac{\omega}{(k_1)_x} = \frac{c_1}{\cos(\theta_{inc})}$ owing to the continuity of the wave on the boundary. Hence, the penetrating wave propagates at a lower speed than that of the sediment, and its propagation speed depends on the incident angle. In a relevant study [14], when modeling the penetrating wave, it was observed that the wave propagation speeds in the sediment deviated from the measured sediment sound speed for incident waves below critical angles. Subsequently, they were fitted to enhance the agreement between the model results and measured acoustic data. The fitted sound speeds in the sediment were 1650, 1685, and 1720 m/s for grazing angles of 22.0°, 24.8°, and 27.2°, respectively, and the sound speeds of water and the sediment were 1530 and 1720 m/s, respectively. The fitted sound speeds are similar to the horizontally propagating sound speed of $\frac{c_1}{\cos(\theta_{inc})}$, which are 1650.2, 1685.4, and 1720.2 m/s

at 22.0°, 24.8°, and 27.2°, respectively. Therefore, they can be explained simply based on the continuity of the wave on the boundary.

The FEM was used to confirm the features of the penetrating wave, which are shown in Figure 2. The numerical environment was the same as that in Figure 1 except for the absence of the target. The penetrating waves were simulated for incident waves with two different incident angles (10° and 20°) at 3 kHz. As expected, the wavefronts in the sediment were vertical, and the sounds propagated horizontally. Each penetrating wave propagated at a distance of the corresponding single wavelength in the sediment during one period, and the propagation speed is calculated by dividing the wavelength in the FEM result with one period. Here, the penetrating wave along the horizontal axis at the fixed depth of 0.02 m was used to measure the wavelength required for calculating the propagation speed. The penetrating wave speeds from the FEM were 1525.6 and 1597.4 m/s for incident angles of 10° and 20° at 3 kHz, respectively. They are similar to those computed using $\frac{c_1}{\cos(\theta_{inc})}$, which are 1523.1 and 1596.3 m/s at 10° and 20°, respectively.

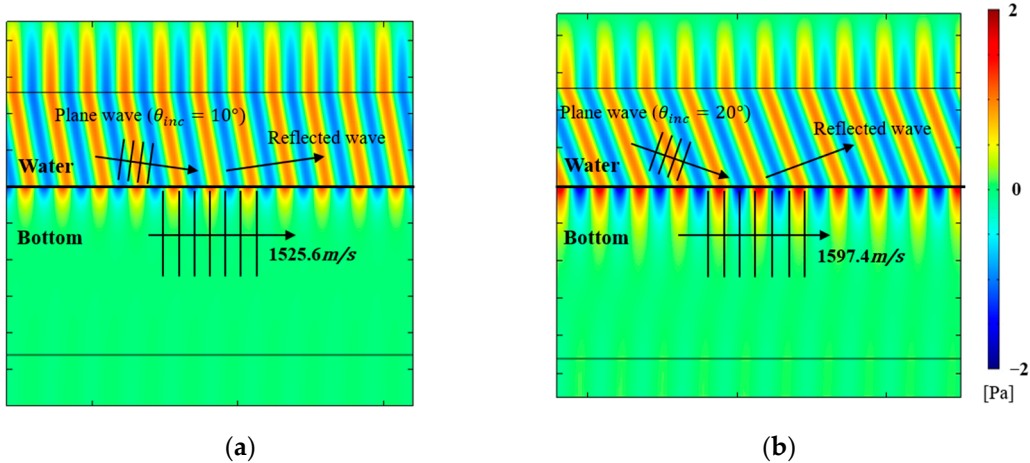

**Figure 2.** Penetrating waves simulated using FEM for incident plane waves with incident angles of (**a**) 10° and (**b**) 20°. The penetrating waves propagates horizontally in the sediment with sound speeds depending on the incident angles and diminish with an increment of depth.

Meanwhile, the plane wave amplitude diminished as the depth increased. To calculate the decaying rate in the FEM, the penetrating wave along the vertical axis at the fixed horizontal range of 0 m was used. Specifically, pressures of the penetrating wave at different depths from the FEM were used, based on Equation (1). The decaying rates using the FEM results were 4.76 and 2.98 for the incident angles of 10° and 20°, respectively, and they are similar to those obtained from the analytical expression of $\alpha(f, \theta_{inc}; c_1, c_2) = \sqrt{(k_1)_x^2 - k_2^2}$ (4.76 and 2.99 at 10° and 20°, respectively).

### 2.2. Target Strength Based on Subcritical Incident Waves

Sounds were reradiated from a target with different amplitudes along the scattering angles ($\theta_s$) after the target was illuminated by an incoming wave with angle ($\theta_i$), which is the angle incident to the target. Here, the incident and scattering angles were measured from the horizontal axis passing through the target center. The target scattering cross-section $\sigma(\theta_i, \theta_s, f)$ (or its logarithmic measure, TS) was calculated using scattered waves at positions encompassing the target, as follows [20]:

$$\sigma(\theta_i, \theta_s, f) = \frac{P_s^2(r, \theta_i, \theta_s, f)r}{P_i^2} \tag{2}$$

where $P_i$ and $P_s$ are the amplitudes of the incoming and scattered waves, respectively; $r$ is the distance between the receiver and target. Because the 2-D target scattering was

simulated for efficiency, the target scattering cross-section was independent of the azimuthal angle, and the square of the distance in the 3-D target scattering was replaced with the distance.

Generally, when calculating the target scattering cross-section, an incident wave with a planar wavefront, whose amplitude is constant (referred to as standard plane wave herein), illuminates the free-field target with an arbitrary incident angle, and the physical properties (such as the density and sound speed) of the medium surrounding the target is used.

Meanwhile, the long-range buried target is illuminated by the penetrating wave below the critical angle, which induces a different target scattering cross-section from the general one. The penetrating wave (the incoming wave for the buried target) propagates horizontally, and the target scattering cross-section becomes independent of the incident angle of the incoming wave to the target; here, $\theta_i = 0$. Furthermore, the incoming wave propagates at a lower speed of $\frac{c_1}{\cos(\theta_{inc})}$ than the original sound speed of the sediment surrounding the target, and its amplitude decays along the depth.

When calculating the target scattering cross-section for the long-range buried target, the features of the incoming wave must be considered; hence, the target scattering cross-section is a function of $\theta_{inc}$ instead of $\theta_i$. Accordingly, the general definition of the target scattering cross-section is replaced with the following equation:

$$\sigma(\theta_{inc}, \theta_s, f) = \frac{P_s^2(r, \theta_{inc}, \theta_s, f)r}{\overline{P}_i^2} \tag{3}$$

Here, $\overline{P}_i$ is the average amplitude of the incoming wave over the target.

$$\overline{P}_i = \sqrt{\frac{1}{2a} \int_{z_c-a}^{z_c+a} A_b^2 e^{-2\alpha z} dz} \tag{4}$$

where $A_b$ is the amplitude of the incoming wave at the depth $z = 0$, $z_c$ is the depth of the target center measured from the water–sediment interface, and $a$ is the radius of the target. Because the target scattering cross-section is the ratio between incoming wave intensity and scattered wave power, it is independent on $A_b$ and $z_c$. $P_s$ is the scattered wave amplitude at a distance of $r$ from the target center when the incoming wave into the free-field target is the evanescent wave having the same decaying rate and propagation speed as the penetrating wave at subcritical angle $\theta_{inc}$. Whereas $P_s$ can be expressed analytically under the 2-D cylindrical shell illuminated by the standard plane wave, the FEM is essential to computing $P_s$ for the incoming evanescent wave.

When calculating a scattered signal by target, a scattered amplitude $s$ is used and has a relationship with the scattering cross section as $\sigma = |s|^2$. Specifically, it is defined for the evanescent wave as follows:

$$s(\theta_{inc}, \theta_s, f) = \frac{p_s(r, \theta_{inc}, \theta_s, f)\sqrt{r}exp(-ikr)}{\overline{P}_i} \tag{5}$$

$p_s$ is a scattered signal and its amplitude is $P_s$. $k$ is the wavenumber of medium surrounding the target and is equivalent to the sediment wavenumber $k_2$ for the buried target. $\sqrt{r}$ and $exp(-ikr)$ compensate spreading loss and phase change during propagation from the target to receiver, respectively.

In this study, we calculated TSs at low (3 kHz) and high (9 kHz) frequencies for two different incoming waves to the target using the FEM; one for the evanescent wave having the same decaying rate and propagation speed as the penetrating wave at the incident angle of 20°, and the other for the standard plane wave in the front of the target. In the TS calculation, the free-field target was surrounded by the same sediment as that in Figure 1. The mesh size for each TS calculation was the one-twelfth acoustic wavelength at the corresponding frequency. By comparing the TSs, the TS features of the buried target illuminated by the subcritical penetrating wave were investigated.

Figure 3 shows the TSs, where Equations (2) and (3) were used for the standard plane and evanescent waves, respectively. $P_s$ (or $p_s$) for calculating the TS was computed along $\theta_s$ from 0° and 360° at the fixed distance ($r = 100$ m). Whereas the standard waves with an incident angle of zero ($\theta_i = 0$) rendered the TSs symmetric, those for the evanescent waves became asymmetric owing to the depth-dependent amplitude. In particular, larger amplitudes of evanescent waves at the upper section of the target yielded larger TSs between 0° and 180°, which manifested primarily at the higher frequency owing to the larger decay rate. The greater TSs in the upper direction by the evanescent waves facilitated the detection of the buried target. However, as the burial depth increased, the scattered signal was weakened by decreasing $\overline{P}_i$.

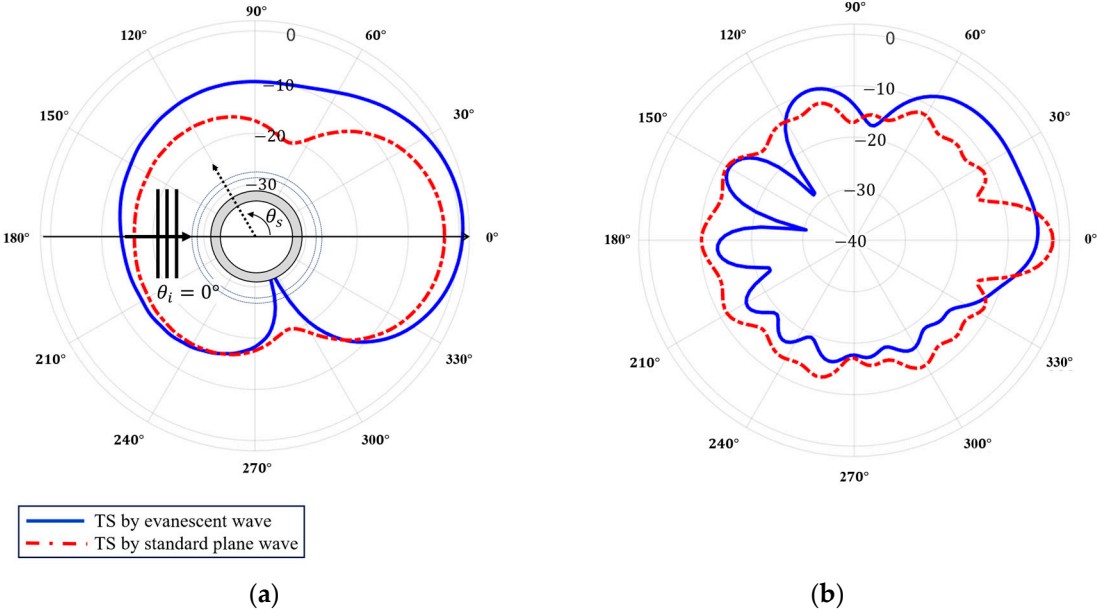

**Figure 3.** Target strengths (in the unit of dB) calculated at (**a**) 3 kHz and (**b**) 9 kHz using FEM when the incoming waves are the evanescent waves having the same decaying rates and propagation speeds as the penetrating waves at subcritical angles; they are compared with those for standard plane waves at corresponding frequencies. In the TS calculation, the free-field target is surrounded by the sediment.

Meanwhile, the TSs for the evanescent waves exhibited more complicated patterns owing to the higher wavenumber induced by the lower sound speed, compared with those for the evanescent and standard plane waves propagating with the bottom sound speed. The features of the evanescent waves make the corresponding TSs distinguished from those of the standard waves, and, thus, scattering patterns of buried target illuminated by subcritical penetrating waves are different from the general ones.

### 2.3. Ray Approach for Modeling Scattered Signals from Long-Range Buried Target

The ray approach has been used to model scattered signals from a target owing to its efficiency. In the ray approach, signals scattered from the target are simulated by combining wave propagations from the source to the target and from the target to the receiver with the target scattering amplitude. Hence, in ray-based modeling, the scattered signals can be analyzed based on their characteristics, such as propagation and scattering, thereby enabling their physical interpretation.

Conventionally, the ray approach has been applied to simulate scattered signals from the target above the sea bottom (proud target) by considering the interference of multiple ray paths from the sea bottom [12,13]. In this study, the ray approach was modified to model scattered signals from long-range buried targets at low-to-high frequencies. It is noteworthy that the bottom roughness was considered for modeling the target scattering at

high frequencies, which enabled an incident wave to penetrate the bottom and illuminate the target [18–20,22]. However, in the current study, with the assumption of a flat bottom, the buried target scattering at high frequency was simulated by considering the features of the penetrating waves below the critical angle and the corresponding scattering amplitude.

Simulating subcritical scattered signals from a buried target is composed of two steps in the ray approach: (1) The subcritical scattering from the buried target and (2) the sound propagating from source to target and from target to receiver. In the first step, the free-field target is surrounded by sediment and is illuminated by the evanescent wave having the same decaying rate and propagation speed as the penetrating waves at a subcritical angle. Subsequently, a sound is reradiated from the target with different amplitudes according to scattering angles (target scattering amplitude or cross-section) and scattered signals arrive at the receiver via single or multiple paths depending on a sound propagating environment.

In this study, for convenience, a receiver for capturing the scattered signal was installed in the bottom, as shown in Figure 4a. The dominant scattering signals propagated via direct and bottom reflected paths, where the cylindrical spreading loss in the bottom indicated a constant sound speed. When using the ray approach, the signal scattered to the receiver in the bottom ($p_s^B$) is denoted as follows:

$$p_s^B = p_1 + p_2 \tag{6}$$

$$p_1 = \overline{P}_i s(\theta_{inc}, \theta_{s1}, f) \frac{exp(ik_2 r_1)}{\sqrt{r_1}} \tag{7}$$

$$p_2 = \overline{P}_i s(\theta_{inc}, \theta_{s2}, f) R(\theta'_{s2}) \frac{exp(ik_2 r_2)}{\sqrt{r_2}} \tag{8}$$

$A_b$ in $\overline{P}_i$ based on Equation (4) is $A_w \check{T}(\theta_{inc})$ where $A_w$ and $\check{T}$ are the incident wave amplitude and the transmission coefficient from the water to the bottom, respectively. $R$ is the reflection coefficient from the bottom to water. The scattering amplitude $s$ is calculated using scattered signals from the free-field target illuminated by the evanescent wave having the same decaying rate and propagation speed as the subcritical penetrating wave, based on Equation (5). $\theta_{s1,s2}$ and $r_{1,2}$ are the scattering angles and propagation distances for the direct and bottom reflected paths, respectively (Figure 4a). $\theta'_{s2}$ is expressed differently for forward and backward scatterings: $\theta'_{s2} = \theta_{s2}$ for forward scattering and $\theta'_{s2} = \pi - \theta_{s2}$ for backward scattering.

A scattered signal was simulated using the ray approach for a receiver at the same depth of the target center (0.35 m) under the numerical environment of Figure 1. The distance between receiver and target is set to 2 m. Subsequently, the signal was compared with that obtained using the FEM with respect to amplitude (sound pressure level) and phase, as shown in Figure 4b. The incident wave from the water, whose amplitude was 1 Pa, was incident into the bottom with the angle of 20° (i.e., $A_w = 1$ Pa and $\theta_{inc} = 20°$). While overall patterns such as peak positions were similar between the scattered signals, discrepancies were observed owing to approximation in the ray approach. Whereas the FEM fully solved the Helmholtz equation with the given boundary conditions, the ray approach derived the scattered signal approximately by combining wave propagation with target scattering pattern (i.e., target scattering amplitude). Particularly, in the ray approach, the target was treated as a directional point source by non-uniform reradiation from the scattering amplitude, which accounted for the difference of the ray result from the FEM result. The discrepancies were more apparent when the receiver was located closer to the target owing to the complicated near-field target scattering [5,23]. However, as shown in Figure 4b, the ray approach can provide reliable scattering results efficiently in the frequency range of 100 Hz–15 kHz in a unified manner, particularly, for the far-field receiver from the target.

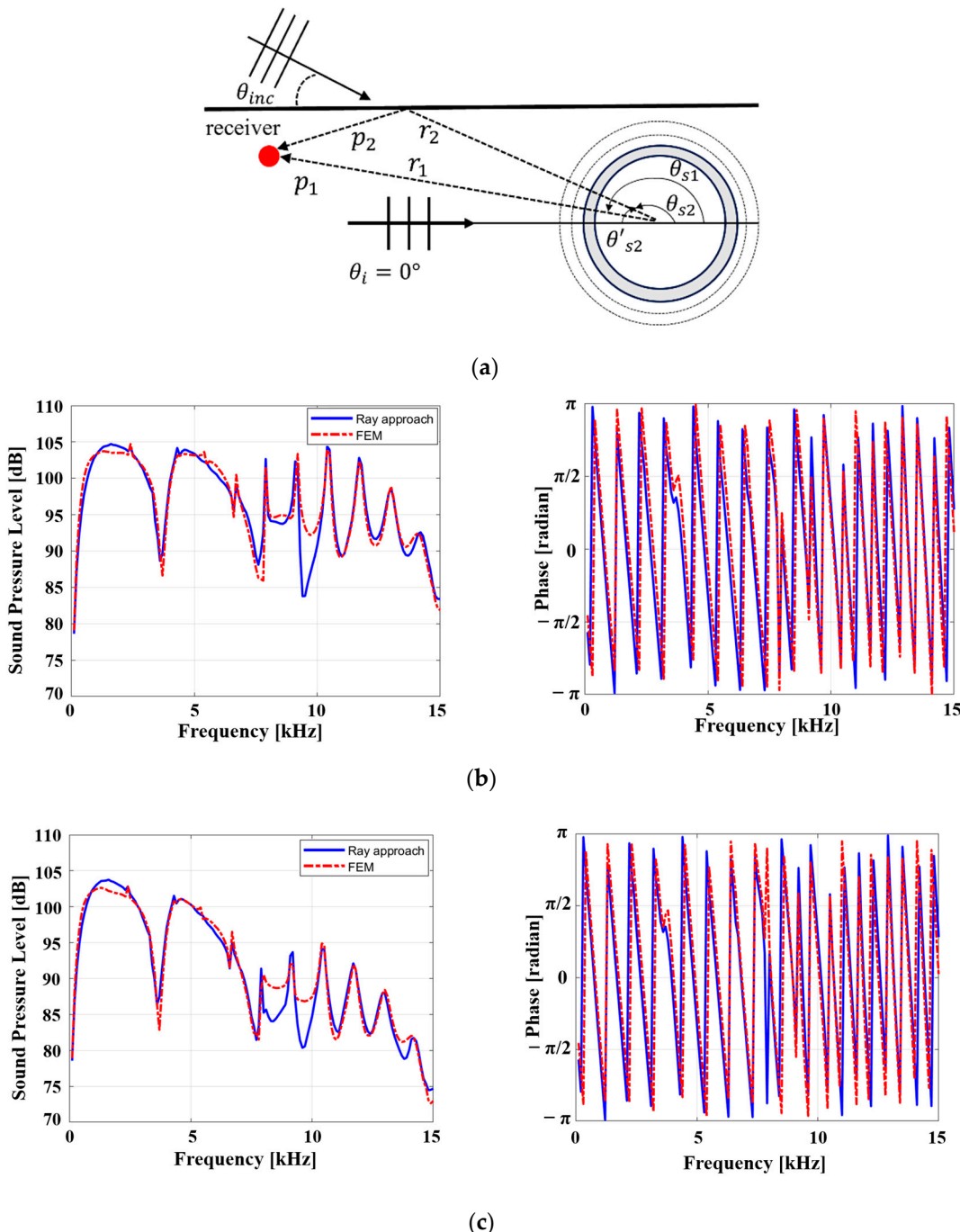

**Figure 4.** (**a**) Dominant scattered paths for buried target (direct and bottom reflected paths). Comparison of simulated scattered signals between ray approach and FEM for low-to-high frequencies with respect to amplitude (sound pressure level) and phase when (**b**) neglecting and (**c**) considering bottom attenuation.

While peak amplitudes were less than those yielded by the shell target illuminated by the standard plane wave owing to the larger decaying rate of penetrating wave at higher frequencies, the strong peaks were still yielded by the MFE from the buried shell target.

So far, bottom attenuation was neglected to focus on the effects of penetrating waves on subcritical target scattering. To inspect the MFE detectability in a more realistic ocean environment, the scattered signals were simulated using the FEM and ray approach while considering the bottom attenuation $\alpha^{(\lambda)} = 0.5$ dB/wavelength, and they were in good agreement (Figure 4c). In the ray approach, the scattering amplitude was calculated using

the free-field target surrounded by the bottom with attenuation and the additional terms of $e^{+\alpha r}$, $e^{-\alpha r_1}$, and $e^{-\alpha r_2}$ were added to Equations (5), (7) and (8) to account for loss by the bottom attenuation ($\alpha = \alpha^{(\lambda)}/8.686\lambda$, where $\lambda$ is a wavelength) [24]. The MFE peaks were observed although the high-frequency peaks diminished owing to the significant bottom attenuation at high frequency.

Finally, the frequency-domain solutions from the FEM and ray approach in Figure 4b,c were converted to the corresponding time signals using Fourier transform as shown in Figure 5; 1–14 kHz linear frequency modulated pulse signal was used as a source waveform. While fictitious signals from numerical errors during the conversion were observed in the first and later parts, the time signals from the FEM and ray approach were in good agreement (e.g., the same first arrival time near 1.2 ms).

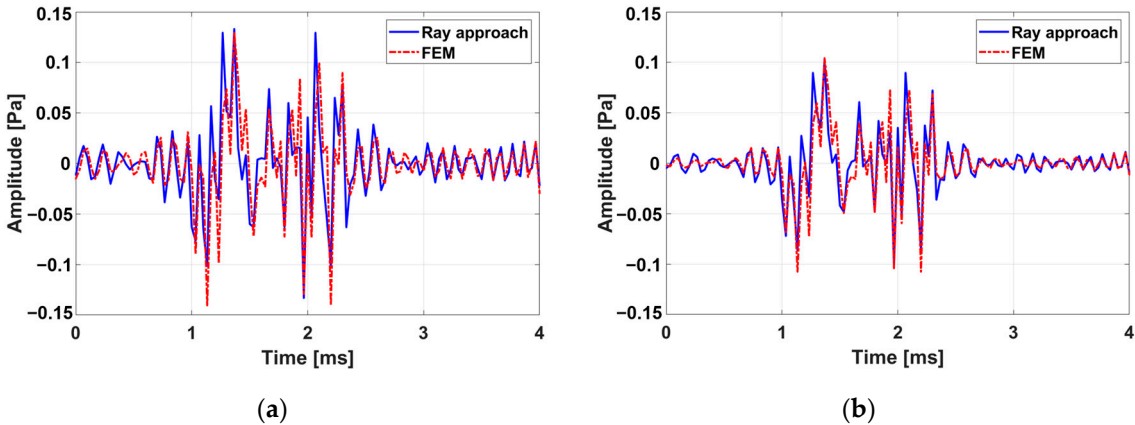

(**a**)                    (**b**)

**Figure 5.** Comparison of simulated scattered signals between ray approach and FEM in time domain when (**a**) neglecting and (**b**) considering bottom attenuation. A 1–14 kHz linear frequency modulated pulse signal is used as a source waveform.

## 3. Discussion

In previous studies, a buried target was presumed to be not illuminated by a highly decaying penetrating wave, and the subcritical target scattering at high frequency was explained based on the bottom roughness [18–20]. In particular, using a monostatic sonar system, where the source and receiver are collocated, it is difficult to exploit the MFE to detect the buried target based on the bottom attenuation along with the transmission loss from the bottom to water. However, Figure 4b,c show the possibility of detecting the buried target under conditions of distant source and flat bottom using a bistatic sonar with a broadband source signal, which includes low-to-mid frequencies.

When the receiver approaches the buried target near the water–sediment interface, the scattered MFE signals can be observed using the broadband source signal. To inspect the properties of bistatic sonar, the scattered signal at the receiver in water ($p_s^W$) was calculated using the ray approach. In fact, $p_s^W$ has a single ray path (eigenray trajectory) that accounts for the refraction by the sound speed difference between water and bottom; $p_s^W$ is expressed as (further details are shown in the Appendix A)

$$p_s^W = \overline{P}_i s(\theta_{inc}, \theta_1, f)\hat{T}(\theta_1')\sqrt{\frac{\rho_1 c_1}{\rho_2 c_2}\frac{1}{\sin(\theta_2')}\frac{d\theta_1'}{dr_h}}, \qquad (9)$$

where $\rho_{1,2}$, and $c_{1,2}$ are the densities and sound speeds at water and bottom, respectively; $\theta_1'$ and $\theta_2'$ are associated with the departure angle from the target ($\theta_1$) and arrival angle ($\theta_2$) at the receiver, respectively; $\theta_{1,2}' = \theta_{1,2}$ for forward scattering, and $\theta_{1,2}' = \pi - \theta_{1,2}$ for backward scattering; $r_h$ is the horizontal distance between the target and receiver; $\overline{P}_i$ is the same as that in Equations (7) and (8). The propagation loss in the 2-D refractive medium is considered by the last term, whose derivative can be obtained using a ray-based

propagation model [21,24], and a transmission coefficient from bottom to water ($\hat{T}$) is added. The terms before and after *s* account for sound propagation before and after scattering, respectively.

The scattered signals at the receivers in water were simulated based on Equation (9) using the same incident wave as that in Figure 4 when the receivers moving along the horizontal range from −50 to 50 m at a fixed depth of 10 m from the bottom without attenuation (Figure 6a). Although MFEs were indicated in the scattered signals at the receivers far from the target, they were reduced by significant transmission loss from the bottom to water at a low grazing angle into water. In a real ocean environment, it is difficult to exploit the MFEs using monostatic sonar far from the target owing to the transmission loss and bottom attenuation. However, as the receiver approached the target, a slight transmission loss at high grazing angles resulted in conspicuous scattered signals that appeared broadly around the peaks, owing to the MFEs (more pronounced in positive distances corresponding to forward scattering directions).

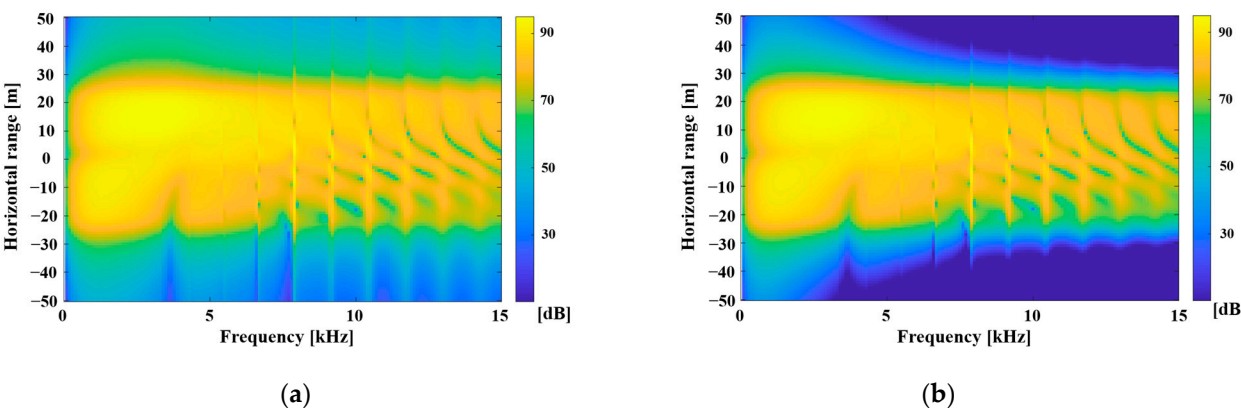

(**a**)　　　　　　　　　　　　　　　　　　　　　　　　　　　　(**b**)

**Figure 6.** Scattered signals of moving receivers along horizontal range from −50 to 50 m at fixed depth of 10 m from the bottom (**a**) without and (**b**) with attenuation.

To inspect the MFE detectability in a more realistic ocean environment, the scattered signals were simulated while considering the bottom attenuation (0.5 dB/wavelength). As shown in Figure 6b, the high frequency scattered signals at the receivers far from the target were considerably weakened by the long ray travel distance with significant attenuation at the high frequency in the bottom. However, the closer scattered signals experienced less loss by the shorter propagation distance at the bottom. Hence, to improve the detection performance of the buried target, a bistatic sonar with a broadband source signal that induces MFE is preferred.

## 4. Conclusions

In this study, scattered signals from long-range buried targets were simulated using the ray approach by considering the features of penetrating waves below the critical angle and corresponding the target scattering cross-section. In particular, the penetrating waves, which decayed along the depth and propagated at speeds lower than the sound speed of the bottom, distinguished the target scattering cross-sections from those of standard waves. Furthermore, they exhibited asymmetric patterns (larger amplitudes in the upper directions) owing to the decaying wave, even when incoming waves were propagating from the front of the symmetric target. Additionally, they became more complicated owing to the higher wavenumbers arising from the lower speeds.

Conventionally, scattered signals from long-range buried targets are simulated using the FEM, which incurs a significant computational cost. Hence, its application is restricted to simulating scattered signals from small targets or scattered signals below high-frequency regions. However, in the ray approach, once a target scattering cross-section (or target scattering amplitude) is computed via the FEM, fields scattered by the target are simulated

efficiently in all frequency ranges without necessitating significant computational cost and memory.

Furthermore, the scattered signals can be analyzed by decomposing the effects of the propagation and target scattering cross-section via the ray approach. MFE was indicated at subcritical scattering from the long-range buried target, which resulted in intensive peaks, as shown in Figures 4 and 5. In particular, a buried target (in particular, flush-buried target) was detected via the MFE using receivers near the target, with less propagation loss from the target to the receiver.

In this study, a 2-D target was used to efficiently investigate the long-range buried target scattering and physical phenomena involved in the target scattering. The full FEM is difficult to apply to 3-D target scattering at high frequencies due to computational cost and memory. However, the ray approach can efficiently simulate scattered signals from the 3-D target (in particular, axis-symmetric targets whose scattering amplitudes can be calculated at a low computational cost [9]).

**Author Contributions:** Conceptualization, Y.-S.C. and Y.C.; methodology, Y.-S.C. and Y.C.; software, Y.-S.C. and G.C.; validation, Y.-S.C., K.L. and Y.C.; formal analysis, Y.-S.C., S.-H.B. and Y.C.; writing—original draft preparation, Y.-S.C., G.C. and Y.C.; writing—review and editing, S.-H.B. and K.L.; All authors have read and agreed to the published version of the manuscript.

**Funding:** This research was supported by Korea Research Institute of Ships and Ocean engineering a grand from Endowment Project of "Development of core technology for synthetic aperture sonar using mid-and low frequency moving sensor array" funded by Ministry of Oceans and Fisheries (PES4380). This research was supported by AUV Fleet and its Operation System Development for Quick Response of Search on Marine Disasters of Korea Institute of Marine Science & Technology Promotion (KIMST) funded by the Korea Coast Guard Agency (KIMST-20210547).

**Institutional Review Board Statement:** Not applicable.

**Informed Consent Statement:** Not applicable.

**Data Availability Statement:** Not applicable.

**Conflicts of Interest:** The authors declare no conflict of interest.

## Appendix A. Propagation Loss of Refracted Ray in 2-D Space

When a sound emitted from a source is refracted by sound speed variation in the medium (i.e., a ray with departure angle $\theta_s$ reaches a receiver with arrival angle $\theta_r$, by Snell's law), its propagation loss is different from that in a medium with a constant sound speed. The law of energy conservation can be used to derive the propagation loss in the 2-D refractive medium as in Refs. [21,24]. In other words, the acoustic power from the source, $dW_s = \frac{p_s^2 d\Omega_s}{\rho_s c_s}$, must be equal to the acoustic power, $dW_r = \frac{p_r^2 d\Omega_r}{\rho_r c_r}$, where $p_{s,r}$, $\rho_{s,r}$, and $c_{s,r}$ are the pressure, density, and sound speed at a reference distance of 1 m from the source and receiver, respectively. Furthermore, $d\Omega_s = d\theta_s$ and $d\Omega_r = dr_h \sin(\theta_r)$, where $dr_h$ is the horizontal range difference at the receiver based on the departure angle variation $d\theta_s$. The ratio of the pressures, which indicates the propagation loss, is denoted as $\frac{p_r}{p_s} = \sqrt{\frac{\rho_r c_r}{\rho_s c_s} \frac{1}{\sin(\theta_r)} \frac{d\theta_s}{dr_h}}$.

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
