# Peer review of "Ray-Based Analysis of Subcritical Scattering from Buried Target"

_jmse, doi:10.3390/jmse11020307_

Round 1

Reviewer 1 Report

Interesting study on the use of bistatic sonar coupled with ray-based analysis for detecting buried objects in a seabed. This research is very relevant to detecting man-made targets (such as mines) buried in the seabed. Overall, the work is well presented but needs some improvements. 

1. The use of bistatic sonar involves embedding receivers close to the target. How would you quantify the number of receivers required for detecting a small target over a large area of seabed?

2. Additional figures and explanations could be presented to show how wave scattering would be affected for different distances between the target and the receiver. Figures showing differences in sound pressure levels (for example) for different distances between the target and the receiver could be included. How would the figures change in the presence and absence of a target?

3. How would the irregularity of the seabed affect the target detection capabilities of the system?

4. How would the shape of the target affect detectability? 

5. How would you differentiate between direct travel from the source to the receiver vs scattering from the target to the receiver? 

6. Please include more details about parameters that need to be considered for the broadband source for different depths of the water column above the seabed. 

7. Some graphical representations (figures) explaining the ray-based analysis concept could be incorporated. 

8. Some experimental verification of the simulated results would be interesting. 

Author Response

Please, see the attachment including responses to the comments.

Reviewer 2 Report

Please check spelling; for instance, change "filled with the air." to "filled with air." (line 106). Indicate explicitly that bottom shear properties are not being taken into account. How fast are ray calculations relative to FEM calculations? Ray calculations are 2-D, why the reference to 3-D calculations? Is Eq(1) defined by the authors or is it a general definition? If general, include a reference. The calculations of Sound Pressure Level require a value of Sound Level, what value has been chosen for the calculations? Why?

Author Response

(The authors gave the same response as above.)

Reviewer 3 Report

Kindly see the attached file, thank you.

Author Response

(The authors gave the same response as above.)

Reviewer 4 Report

I appreciated a lot the paper. I suggest to make a more quantitative comparison between the proposed technique and the reference one, giving quantitatvive differences and discuss it.

Author Response

(The authors gave the same response as above.)

Reviewer 5 Report

1. How critical angle is manage to detect the target?

2. What is threshold to detect amplitude?

3. Add some latest article in literature review

4. Rewrite abstract and conclusion 

Author Response

(The authors gave the same response as above.)

Round 2

Reviewer 1 Report

Thank you for responding to the comments. Excellent work overall. I look forward to future experimental work. 

Reviewer 5 Report

Authors incorporated all the changes in the article